# Epidemiology of Malnutrition among Children with Cerebral Palsy in Low- and Middle-Income Countries: Findings from the Global LMIC CP Register

**DOI:** 10.3390/nu13113676

**Published:** 2021-10-20

**Authors:** Israt Jahan, Mohammad Muhit, Denny Hardianto, Francis Laryea, Samuel Kofi Amponsah, Amir Banjara Chhetri, Hayley Smithers-Sheedy, Sarah McIntyre, Nadia Badawi, Gulam Khandaker

**Affiliations:** 1CSF Global, Dhaka 1213, Bangladesh; israt.jahan@cqumail.com (I.J.); mmuhit@hotmail.com (M.M.); 2Asian Institute of Disability and Development (AIDD), University of South Asia, Dhaka 1212, Bangladesh; 3School of Health, Medical and Applied Sciences, Central Queensland University, Rockhampton, QLD 4701, Australia; 4CSF Global Indonesia, Waikabubak 87214, Indonesia; dennydrgmph@gmail.com; 5Korle bu Teaching Hospital, Accra 77, Ghana; francislaryea.fl@gmail.com; 6Health Information Department, Christian Health Association of Ghana, Accra AN 7316, Ghana; skamponsah001@st.ug.edu.gh; 7CSF Global Nepal, Balaju, Kathmandu 44611, Nepal; amirbanjara@yahoo.com; 8Cerebral Palsy Alliance, Sydney Medical School, The University of Sydney, Camperdown, NSW 2086, Australia; HSmithersSheedy@cerebralpalsy.org.au (H.S.-S.); SMcIntyre@cerebralpalsy.org.au (S.M.); nadia.badawi@health.nsw.gov.au (N.B.); 9Grace Centre for Newborn Intensive Care, Sydney Children’s Hospital Network, Westmead, NSW 2145, Australia; 10Discipline of Child and Adolescent Health, Sydney Medical School, The University of Sydney, Sydney, NSW 2006, Australia; 11Central Queensland Public Health Unit, Central Queensland Hospital and Health Service, Rockhampton, QLD 4700, Australia

**Keywords:** malnutrition, underweight, stunting, cerebral palsy, children, LMIC, disability, register

## Abstract

Background: This study aimed to describe the epidemiology of malnutrition among children with cerebral palsy (CP) in low- and middle-income countries (LMICs). Methods: Data from children with confirmed CP aged <18 years registered into the Global LMIC CP Register (GLM CPR) from Bangladesh, Indonesia, Nepal, and Ghana were included. Anthropometric measurements were collected, and nutritional status was determined following the WHO guidelines. Descriptive statistics and adjusted logistic regression were used to describe the nutritional status and identify predictors of malnutrition. Results: Between January 2015 and December 2020, 3619 children with CP were registered into the GLM CPR (median age at assessment: 7.0 years, 39% female). Overall, 72–98% of children from Bangladesh, Indonesia, Nepal, and Ghana had at least one form of undernutrition. The adjusted analysis showed, older age, low maternal education, spastic tri/quadriplegia, and Gross Motor Functional Classification System (GMFCS) levels III–V were significant predictors of underweight and stunting among children with CP in Bangladesh. In Nepal, female children, GMFCS III–V had higher odds of underweight and stunting. In Ghana, low maternal education was significantly associated with underweight, whereas older age and the presence of associated impairments were the significant predictors of stunting among children with CP. Having a GMFCS of III–V increased the odds of being underweight among children in Indonesia; however, no predictors were identified for stunting, as nearly all children with CP registered from Indonesia were stunted. Conclusion: Most children with CP in GLM CPR had undernutrition. Maternal education and moderate-to-severe motor impairment (GMFCS III–V) were significant predictors. Practical nutrition education to mothers/caregivers and management guidelines according to the motor severity using local resources could improve the nutritional outcome of children with CP in LMICs.

## 1. Introduction

According to the Global Burden of Disease study, 94% of the global burden of childhood disabilities occur in low- and middle-income countries (LMICs) [1]. Although the estimates excluded neurodevelopmental disabilities such as cerebral palsy (CP), recent advancements in epidemiological evidence from LMICs such as Bangladesh, Uganda, Indonesia, Nepal, and Ghana also suggest that the burden of CP is high in LMICs and are mostly due to preventable causes [2,3,4]. CP not only affects children’s gross and fine motor function but can also have impact on swallowing, feeding, and digestion, resulting in malnutrition [5,6].

Children with CP are at increased risk of malnutrition and growth failure [5,6]. Data from different LMICs have repeatedly identified the high burden of moderate-to-severe undernutrition among children with CP in LMICs [7,8,9,10]. The consequences of malnutrition in this vulnerable group are diverse. Malnutrition increases susceptibility to infection, reduces cardiac output, increases hospitalization, reduces social participation and quality of life, and lowers the probability of survival [5,6,11,12,13]. Nutrition intervention is therefore crucial and should be included in planning/designing an intervention package for children with CP. The lack of adequate resources to facilitate sufficient and need-based customized intervention for children with CP is a constant challenge, particularly in LMIC settings. Generating comparable epidemiological data from the LMICs on these crucial issues (e.g., epidemiology of malnutrition among children with CP) could guide the development of strategies that are adaptable and replicable in similar settings.

In 2018, the global LMIC CP register (GLM CPR) was established to provide infrastructural support to new/established CP registers in LMICs with the aim of generating robust and comparable data on epidemiology of CP in LMICs [2]. The collaborative framework also aims to provide a platform to facilitate multi-country translational research for better outcomes of children with CP in LMICs [2]. Currently, four countries (Bangladesh, Indonesia, Nepal, and Ghana) are contributing data to the GLM CPR. Data sharing is in progress with a few other LMICs [2]. The latest publications from the GLM CPR have identified some commonalities, as well as some regional/geographic differences, in etiology, clinical severity, rehabilitation, educational status, and causes behind low service utilizations among children with CP in LMICs [2,14]. While such data are crucial and could act as the baseline for evidence-based planning and the implementation of interventions, comparable data illustrating the nutritional status of children with CP is lacking in LMICs. This study aimed to describe the epidemiology of malnutrition among children with CP in LMICs using data from the GLM CPR.

## 2. Materials and Methods

We used data from the Global LMIC CP Register (GLM CPR).

### 2.1. Participants and Settings

Children with clinically confirmed CP aged <18 years were registered in their respective country-specific CP registers of the GLM CPR following a detailed neurodevelopmental/medical assessment.

Deidentified data was collated from population/community-based registers/study in Bangladesh (i.e., Bangladesh CP Register–BCPR, established in 2015), Indonesia (i.e., community-based survey of children with CP in Sumba in 2017), Nepal (i.e., Nepal CP Register–NCPR, established in 2018), and institution-based register in Ghana (i.e., Ghana CP Register–GCPR, established in 2018), which forms the GLM CPR [2]. The details of the geographical locations and settings are available in previous publications [2,14]. All CP registers included in the GLM CPR used a standard case definition and protocol, thus generating comparable data. The case definition, details about surveillance mechanisms adopted, and the medical assessments are available in previous publications [2,14].

### 2.2. Data Collection

#### 2.2.1. Anthropometric Measurements

The primary outcome measure of this study was the nutritional status of children with CP registered into the GLM CPR. Anthropometric measurements (weight, length/height, and Mid-Upper Arm Circumference (MUAC)) were collected to assess their nutritional status. The detailed methods of the anthropometric measurements are available in Appendix A and were reported in our earlier publications [15].

Three repeated measures were taken for all measurements, and the average was documented.

#### 2.2.2. Sociodemographic and Clinical Characteristics

Data on the agreed core variables are collected using a similar structured template following a standard protocol as part of the GLM CPR. A multidisciplinary medical assessment team (including a pediatrician, physiotherapist, occupational therapist, dietitian, and a counselor) interviewed the primary caregivers of children with confirmed CP in person for detailed sociodemographic and clinical history, completed functional and clinical assessments, and reviewed available medical records of each child following a harmonized protocol. The details about the core variables and data collection process have been summarized in our previous publications [2,14,16]. For this current study/data analysis, the following variables were included. (i) anthropometric measurements, (ii) sociodemographic characteristics, (iii) predominant motor type and topography, (iv) gross motor function classification system (GMFCS) level, (v) presence/absence of any associated impairments, and (vi) presence/absence of dysphagia and gastroesophageal reflux (data were not available for Ghana).

#### 2.2.3. Secondary Data from Demographic and Health Survey (DHS)

In addition to the primary data collected from different LMIC CP Registers, we extracted secondary data from the latest demographic and health surveys (DHS) of the respective LMICs. Available data regarding the anthropometric measurements (weight and height) were extracted for children aged five years or less living in the same region of children with CP registered into the GLM CPR. The following DHS datasets were used: (i) Bangladesh Demographic and Health Survey (BDHS) 2018 (data were extracted for the Rajshahi Division), (ii) Nepal Demographic and Health Survey (NDHS) 2017 (data were extracted for Province 4), and (iii) Ghana Demographic and Health Survey (GDHS) 2014 (data were extracted for the Brong Ahafo and Ashanti regions) [17]. We did not extract any data from the Indonesia DHS since the DHS does not report childhood nutritional status in the country.

### 2.3. Data Management and Analysis

#### 2.3.1. Indicators Used to Assess Nutritional Status

All anthropometric measurements were converted to z-scores using WHO Anthro and WHO AnthroPlus software. While estimating the z-scores, the measures of children with CP were compared to the WHO reference curve based on the general population. Multiple indices were used to determine the nutritional status of children registered in the GLM CPR. The weight-for-height z-score (WHZ) and MUAC-for-age z-score (MUACZ) were calculated for children aged ≤61 months; the weight-for-age z-score (WAZ) was measured for children aged ≤121 months; the height-for-age z-score (HAZ) and BMI-for-age z-score (BAZ) were calculated for all the participating children (i.e., <18 years).

#### 2.3.2. Statistical Analysis

A Shapiro–Wilk test was used to describe the distribution of the continuous variables. Descriptive analyses were carried out to report the country-specific nutritional status of children with CP registered into the GLM CPR. Furthermore, the distribution of WAZ, HAZ, and WHZ of children with CP aged five years or less in Bangladesh, Nepal, and Ghana were compared with the regional data of the general population of similar ages (i.e., WAZ, HAZ, and WHZ of children aged five years or less in their respective regions). The chi-square test and Fisher’s exact test were used to compare the proportional differences between groups. The factors that were found significantly related with the outcome measures in the cross-tabulation were entered into the adjusted models (i.e., adjusted logistic regression). Adjusted odds ratios (aOR) with 95% confidence intervals (95% CI) were reported. Graphical presentations were used to illustrate the findings. A *p*-value < 0.05 was used to determine the significant relationship between two factors. All data analyses were completed using SPSS (IBM Corporation, Chicago, IL, USA) version 26.

### 2.4. Ethical Consideration

This study received ethical approval from the Human Research Ethics Committee of the Central Queensland University in Australia (Ref: 0000022562). Each CP register included in the GLM CPR had ethical approval in their respective country (e.g., Bangladesh, Indonesia, Nepal, and Ghana), as described in our previous publications [2,14].

## 3. Results

Between January 2015 and December 2020, 3619 children with clinically confirmed CP were included in the GLM CPR (Bangladesh: *n* = 2852, Indonesia: *n* = 130, Nepal: *n* = 182, and Ghana: *n* = 455). The median (IQR) age at assessment was 7.0 years (3.7, 11.3) (Bangladesh: 7.1 years (3.7, 11.4), Indonesia: 8.5 years (4.7, 12.4), Nepal: 10.0 years (6.6, 14.1), and Ghana: 4.9 years (2.6, 8.2)) and 39.1% were female (female:male in Bangladesh was 1:1.6, in Indonesia was 1:1.3, in Nepal was 1:1.7, and Ghana was 1:1.4).

### 3.1. Overall Nutritional Status

The overall nutritional status of the participating children is presented in Table 1. Overall, 86% (*n* = 2432/2826), 98% (*n* = 126/129), 72% (*n* = 126/174), and 72% (*n* = 324/449) of children with CP in Bangladesh, Indonesia, Nepal, and Ghana, respectively, had at least one form of undernutrition. Moderate-to-severe underweight ranged between 52 and 89% in the population-based settings (i.e., Bangladesh, Indonesia, and Nepal) and 39% in the institution-based settings (i.e., Ghana). Moderate-to-severe stunting and thinness ranged between 64 and 93% and 29 and 36% in the population-based settings and 51% and 24% in the institution-based settings, respectively. Overall, a small number of children were identified as overweight in all four countries (according to the WAZ: 1%, 1%, 5%, and 4% in Bangladesh, Indonesia, Nepal, and Ghana, respectively; according to the BAZ: 9%, 13%, 8%, and 17% in Bangladesh, Indonesia, Nepal, and Ghana, respectively).

### 3.2. Nutritional Status of Children with CP Aged Five Years or Less

The number of children aged five years or less were *n* = 1015, *n* = 37, *n* = 28, and *n* = 226 in Bangladesh, Indonesia, Nepal, and Ghana, respectively. Among them, 38–81% children were underweight, 61–92% were stunted, and 17–35% were wasted in population-based settings, whereas, in the institution-based register in Ghana, these percentages were 44%, 50%, and 39%, respectively (Appendix A).

The WAZ, HAZ, and WHZ distributions of children with CP aged five years or less in Bangladesh, Nepal, and Ghana were compared to the WAZ, HAZ, and WHZ of the general child population of similar ages living in the same region in Figure 1. The distribution curves for all three indices (i.e., WAZ, HAZ, and WHZ) were wider and had a lower proportion of children with CP within the normal range (i.e. between −2SD and +2SD) compared to the general population in all three countries. The median (IQR) values of WAZ for children with CP and the general population of similar ages were, in Bangladesh: −2.9 (−4.0, −1.7) vs. −1.5 (−2.2, −0.8), in Nepal: −1.5 (−2.9, −0.3) vs. −1.0 (−1.7, −0.4), and in Ghana: −1.5 (−2.9, 0.4) vs. −1.0 (−1.0, 0.0). Similarly, the median (IQR) of HAZ were, in Bangladesh: −3.7 (−5.1, −2.3) vs. −1.3 (−2.0, −0.6), in Nepal: −2.1 (−3.2, 0.0) vs. -1.6 (−2.3, −0.7), and in Ghana: −1.8 (−3.5, 0.1) vs. −1.0 (−2.0, 0.0) for children with CP vs. the general population, respectively. Whereas the median (IQR) of WHZ was, in Bangladesh: −1.1 (−2.6, 0.1) vs. −0.7 (−1.3, 0.05), in Nepal: −0.7 (−1.8, 0.5) vs. −0.3 (−0.9, 0.4), and in Ghana −0.5 (−2.8, 0.9) vs. 0.0 (−1.0, 0.0) among children with CP vs. the general population, respectively. These indicate that children with CP aged five years or less in all those three countries registered in the GLM CPR had a comparatively slower growth than the average growth patterns of general child populations of similar ages living in those same regions.

### 3.3. Factors Related to Underweight, Stunting, and Thinness among Children with CP Registered into the GLM CPR

#### 3.3.1. Sociodemographic Characteristics

Table 2 summarizes the relationship between different sociodemographic factors and the nutritional status of children with CP in LMICs registered into the GLM CPR.

##### Age and Sex of the Child

An increasing trend in underweight, stunting, and thinness was observed among children with CP as their ages increased in all four countries. This change was significant for all three indices in Bangladesh (i.e., *p* = 0.02, *p* < 0.001, and *p* < 0.001 for underweight, stunting, and thinness, respectively) and for stunting and thinness in Ghana (*p* < 0.001 and *p* = 0.001, respectively).

Underweight and thinness were slightly higher among female children than male children in Indonesia (male vs. female: 86%, *n* = 38/44 vs. 94%, *n* = 33/35 underweight, *p* = 0.22; 40%, *n* = 25/62 vs. 43.0%, *n* = 21/49, *p* = 0.79, respectively). A similar pattern was observed for underweight and stunting in Nepal (male vs. female: 45%, *n* = 23/51 vs. 69%, *n* = 22/32 underweight, *p* = 0.03 and 60%, *n* = 62/103 vs. 77%, *n* = 47/61 stunted, *p* = 0.03). However, the opposite was observed for underweight (*p* = 0.09), stunting (*p* = 0.10), and thinness (*p* = 0.02) among children with CP in Ghana.

##### Parental Education, Occupation, and Family Income

All three forms of undernutrition (i.e., underweight, stunting, and thinness) were comparatively lower among children with CP whose mothers and/or fathers had completed higher secondary-level education or above in all four countries. The differences were significant in Bangladesh and Ghana (*p* < 0.05 for underweight and stunting in both countries). We also observed a significant positive relationship between maternal and paternal educational levels in all four countries (*p* < 0.001 for all).

No significant relationship between underweight, stunting, thinness, and parental occupation was observed among children registered in the GLM CPR from Indonesia, Nepal, and Ghana. Thinness was significantly higher among children whose fathers were not involved in income-generating activities in Bangladesh (67%, *n* = 12/18, *p* = 0.01). Although a similar pattern was observed for underweight and stunting among participating children in Bangladesh, the differences were not significant (*p* = 0.06, *p* = 0.55, respectively).

No significant relationship between family income and the nutritional status of children was observed in any of the four countries included into the GLM CPR (*p* > 0.05 for all indices in all four countries).

##### Access to Drinking Water and Sanitation

Access to safe drinking water had no significant influence on the nutritional outcome of children registered in the GLM CPR. However, underweight and stunting were significantly higher among children with CP who had no toilet facility at the households in Bangladesh (83%, *n* = 38/45, *p* = 0.001 and 86%, 56/65, *p* < 0.001 respectively). No significant relationship between undernutrition (i.e., underweight, stunting, wasting) and access to sanitation was observed in other three countries (*p* > 0.05 for all).

#### 3.3.2. Clinical Factors

The nutritional status of the children with CP and their detailed clinical features are presented in Table 3. More than half of the children with spastic CP in all four countries had underweight and/or stunting. Furthermore, the proportion of both underweight and stunting were substantially higher among children with spastic tri/quadriplegia than spastic mono/hemiplegia in all four countries. Similarly, children with GMFCS level III–V had a substantially high level of underweight and stunting in all four countries. We further explored the relationship between the GMFCS level and nutritional status of children using multiple indicators (Appendix A). All forms of moderate-to-severe undernutrition was considerably higher among children with GMFCS levels III–V than children with GMFCS levels I–II in all four countries. The proportion of overweight (BAZ > +2SD) also increased among children with GMFCS levels III–V than children with GMFCS I–II in Bangladesh, Nepal, and Ghana, which indicates the presence of a double burden of malnutrition among children with severe motor impairment in those three countries. Underweight, stunting, and thinness were more common among children with associated impairment (−1) in all four countries. Additionally, children with dysphagia had a slightly higher rate of underweight and stunting, in Bangladesh (80%, *n* = 472/593 underweight and 86%, *n* = 697/814 respectively), Indonesia (93.1%, *n* = 27/29 and 96%, *n* = 45/47 respectively), and Nepal (59%, *n* = 20/34 and 75%, *n* = 42/56 respectively). Similar pattern was observed among children with gastro-esophageal reflux.

### 3.4. Predictors of Undernutrition among Children with CP Registered into the GLM CPR

#### 3.4.1. Underweight

In Bangladesh, when adjusted for other covariates, underweight was significantly associated with child’s age, maternal educational level, spastic topography of CP, GMFCS level, and presence of associated impairment. In Indonesia, when adjusted for GMFCS level, children with CP aged 5–9 years had significantly higher odds of underweight than younger children (i.e., 0–4 years) in the cohort. In Nepal, when adjusted for the sex of the children, children with GMFCS level III-V had significantly higher odds of underweight in the cohort. In Ghana, when adjusted for the GMFCS level and associated impairments, the odds of underweight was significantly higher among children whose mothers had received no formal schooling or received primary education compared to mothers who received higher education (Table 4).

#### 3.4.2. Stunting

In Bangladesh, when adjusted for associated impairments, dysphagia and gastroesophageal reflux, stunting among children with CP was significantly associated with their age, maternal educational level, topography, and GMFCS level. In Nepal, stunting was significantly higher among children with GMFCS levels III–V. In Ghana, when adjusted for parental education and GMFCS level, the odds of stunting significantly increased among children aged 10–14 years and 15 years and above compared to 0–4 years and among children with one or two or more associated impairments than those without any associated impairments (Table 5).

#### 3.4.3. Thinness

In Bangladesh, when adjusted for the parental educational level, GMFCS level, and presence of dysphagia, thinness was significantly higher among older children, children with spastic tri/quadriplegia and children with two or more associated impairments. In Indonesia, no significant predictor of thinness was identified for the registered children. In Nepal, the GMFCS level was the only predictor for thinness; thus, an adjusted analysis could not be performed. In Ghana, the adjusted analysis showed children aged 5–9 years and female children had comparatively lesser chance of thinness compared with children aged 0–4 years and male children, respectively, whereas children whose fathers had received secondary education had significantly higher odds of thinness compared to children whose fathers had received higher secondary education and above. (Table 6).

## 4. Discussion

To the best of our knowledge, this is the first study reporting multi-country comparable data on the nutritional status of children with CP from LMICs. We utilized the GLM CPR platform, which enabled several LMIC-based CP registers to work collaboratively and combine and compare data on different clinical and epidemiological aspects, including the nutritional status of children with CP in those countries [2]. Our findings suggest a substantially high burden of moderate-to-severe undernutrition among children with CP in Bangladesh, Indonesia, Nepal, and Ghana. This is similar to the findings reported in other LMICs [7,8,9]. Undernutrition among children with CP may be due to multiple interlinked factors, including nutritional (e.g., dietary intake, diseases, and feeding difficulties) and non-nutritional factors (e.g., growth hormone deficiency and functional status) [5,6]. Nevertheless, when compared to the national and regional data, we observed an overrepresentation of malnutrition among children with CP aged five years or less compared with their peers in the general population [18,19]. Concerningly, we saw an increasing trend in malnutrition among children with CP with increased ages, highlighting a similar scenario.

We observed an inverse relationship between undernutrition and maternal education among children with CP in Bangladesh and Ghana. These findings are consistent with the national data from their respective countries [18,20]. Maternal education plays a crucial role on child nutrition via improved health knowledge, decision-making, and healthcare utilization [21]. Supporting mothers and primary caregivers by the provision of training on nutritional management could be a potential strategy to improve the growth of children with CP in LMICs. Recent studies evaluating nutrition education interventions in Bangladesh, Ghana, and Tanzania have shown some promising results. Supporting primary caregivers of children with CP with feeding skills and nutritional management also improved the quality of life of children and their caregivers [22,23,24,25].

Consistent with previous research, undernutrition was more prevalent among children with spastic tri/quadriplegia, dyskinetic CP and GMFCS levels III–V [7,8,9,26]. Several underlying factors may have led to growth faltering in those children. We observed a positive relationship between spastic tri/quadriplegia and a severe GMFCS level and associated impairments among children in the GLM CPR. Our study findings, as well as evidence from other studies, also suggest that the risk of feeding difficulties increases in children with increasing motor severity (e.g., a higher GMFCS level) [5,6]. We also observed a significantly high burden of underweight, stunting, and thinness among children with dysphagia and a positive relationship between dysphagia and gastroesophageal reflux among children with CP in Bangladesh. All these factors directly impaired their nutritional intake [6,27]. Additionally, the motor difficulties can also interfere with self-feeding [6]. All these factors can collectively affect food consumption, resulting in growth faltering among children with CP [28]. If those underlying issues remain unresolved, the gap between requirement and intake is likely to increase as the child grows, which could accelerate the growth faltering. In our study, we also observed an increasing trend in the undernutrition rate with increasing ages of the children with CP.

The nutritional management of children with CP should be evidence-based. This should include with careful estimation/consideration of a child’s nutritional requirements, necessary food modifications, and support to resolve or minimize the impact of underlying conditions or environmental issues that impact their dietary intake [6]. Children with more severe disabilities, like the majority in our cohort, could benefit from comprehensive nutrition interventions from a multidisciplinary team to support oral feeding and/or enteral nutrition as required [29,30]. In the resource-constrained settings of LMICs (in terms of trained human resources, infrastructure, technique, and advanced equipment) [31], multidisciplinary customized intervention programs may not be feasible or cost-effective. A recent study from the GLM CPR platform showed a relatively low accessibility to the rehabilitation services among children with CP in LMICs mostly due to a lack of awareness, financial constraints, and transport problems [14]. The shortage of trained professionals, e.g., speech pathologist, dietitian/clinical nutritionists, and gastroenterologists further impedes the proper nutritional management for children with CP in LMICs. Carefully designed oral nutrition intervention programs with a community-based approach however could overcome the barriers and benefit many children with CP in LMICs.

Our study has generated multi-country data on the social and clinical factors affecting the nutritional status of children with CP in LMICs. We observed some variations between the countries in terms of the malnutrition rate and underlying factors/predictors of malnutrition among children with CP. The differences could be related to cultural and social aspects. For instance, the malnutrition rate was substantially high among children with CP in Indonesia compared to others in the GLM CPR cohorts. When explored further, we observed that the study area in Indonesia had a relatively higher rate of food insecurity and lower socioeconomic status compared to the national average, which could be a major contributing factor to such a high burden of malnutrition among those in that cohort [32]. The opposite was observed in Ghana, and the malnutrition rate was comparatively lower among children with CP in Ghana than the other cohorts. This could be due to the surveillance mechanism adopted (i.e., institution-based) in GCPR. A similar pattern was reported in other recent institution-based/hospital-based studies in Argentina and Vietnam [8,9]. Nevertheless, these data could act as the baseline, vital for designing nutrition-specific and nutrition-sensitive interventions for children with CP in LMICs. The GLM CPR data and platform could be used to develop a general and severity-specific nutritional management guideline for children with CP in LMICs. The findings and the platform could be used to conduct multi-country translational research and develop culturally sensitive, feasible, effective, and adaptable nutritional interventions for children with CP in LMICs.

Despite considerable effort, this study had several limitations. (i) Due to the nature of the data, a causal relationship between undernutrition and different factors among children with CP could not be established. (ii) The anthropometric indices used in this study have limitations in reflecting the nutritional status precisely, yet those measurements and indicators are considered feasible assessments for rural settings like our study sites. (iii) It is established that children with CP grow differently than children without CP; thus, comparing their growth with the normative curve/general child population could underestimate their nutritional status and overestimate the burden of malnutrition [33,34,35]. It is therefore recommended to compare their anthropometric measurements with CP-specific growth charts [36,37]. However, in LMICs like ours, no CP-specific growth curves are available to date. Therefore, like many other studies, we had to rely on the WHO growth chart to report the nutritional status of children with CP in LMICs. 

## 5. Conclusions

Despite the high burden of malnutrition among the general population in LMICs, the burden of malnutrition is even higher among children with CP in LMICs when compared with the general population, which is concerning. The study findings have made an important contribution to our understanding of the nutritional status and identification of both the sociodemographic and clinical predictors of malnutrition among children with CP in LMICs. These factors should be considered while designing any health-related interventions for children with CP in LMIC settings. The findings also have implications at different tiers, at the individual and community levels, for service provisions and research, as well as in policymaking. The GLM CPR has enabled the development of a knowledge base, and over time, we expect that the network will grow with other LMICs joining the platform. We also expect the findings and the GLM CPR platform will play a crucial role in the development of common protocols and guidelines for the nutritional management of children with CP in LMICs in the near future and support achieving the sustainable development goal targets.

## Figures and Tables

**Figure 1 nutrients-13-03676-f001:**
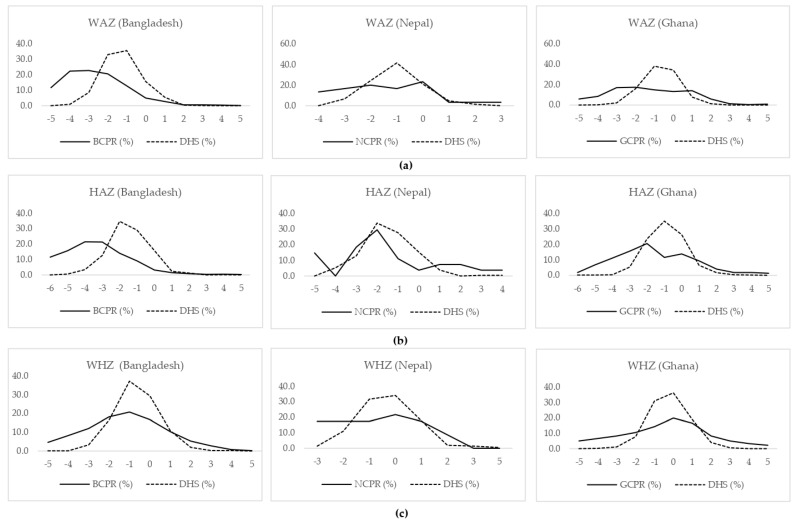
(**a**) WAZ score distribution among children with CP aged ≤5 years vs. their peers from the general population in the same region (DHS data were not available for Indonesia). (**b**) HAZ score distribution among children with CP aged ≤5 years vs. their peers from the general population in the same region (DHS data were not available for Indonesia). (**c**) WHZ score distribution among children with CP aged ≤5 years vs. their peers from the general population in the same region (DHS data were not available for Indonesia).

**Table 1 nutrients-13-03676-t001:** Nutritional status of children with CP registered in the GLM CPR.

Indicator	Weight-for-Age Z Score (WAZ) ^1^	Height-for-Age Z Score (HAZ)	BMI-for-Age Z Score (BAZ)	Weight-for-Height Z Score (WHZ) ^1^	MUAC-for-Age Z Score (MUACZ) ^1^
**Bangladesh**					
*n*	1971	2800	2794	979	1012
Mean (SD)	−3.0 (2.3)	−3.8 (2.9)	−0.8 (3.9)	−1.2 (2.7)	−1.3 (1.7)
Median (IQR)	−3.1 (−4.3, −1.8)	−3.7 (−5.4, −2.1)	−1.2 (−2.6, 0.3)	−1.2 (−2.6, 0.1)	−1.1 (−2.1, −0.3)
Nutritional status, *n* (%)					
Severely undernourished ^2^	1033 (52.4)	1708 (61.0)	575 (20.6)	197 (20.1)	104 (10.3)
Undernourished ^2^	398 (20.2)	453 (16.2)	386 (13.8)	123 (12.6)	159 (15.7)
Normal ^2^	511 (25.9)	600 (21.4)	1589 (56.9)	598 (61.1)	733 (72.4)
Overnourished ^2^	29 (1.5)	39 (1.4)	244 (8.7)	61 (6.2)	16 (1.6)
**Indonesia**					
*n*	80	128	127	32	37
Mean (SD)	−4.1 (2.0)	−5.5 (2.9)	−0.9 (3.3)	−0.8 (2.6)	−1.5 (1.9)
Median (IQR)	−4.2 (−5.0, −3.1)	−5.6 (−7.0, −4.1)	−1.0 (−2.6, 0.8)	−0.4 (−2.2, 1.1)	−1.3 (−2.1, −0.3)
Nutritional status, *n* (%)					
Severely undernourished ^2^	63 (78.8)	110 (85.9)	25 (19.7)	7 (21.9)	6 (16.2)
Undernourished ^2^	8 (10.0)	9 (7.0)	21 (16.5)	1 (3.1)	4 (10.8)
Normal ^2^	8 (10.0)	7 (5.5)	65 (51.2)	18 (56.3)	27 (73.0)
Overnourished ^2^	1 (1.3)	2 (1.6)	16 (12.6)	6 (18.8)	0 (0.0)
**Nepal**					
*n*	87	170	164	26	28
Mean (SD)	−2.2 (1.9)	−2.9 (2.6)	−0.5 (4.1)	−0.5 (1.6)	−0.9 (1.4)
Median (IQR)	−2.1 (−3.8, −0.9)	−2.8 (−4.5, −1.4)	−1.1 (−2.5, 0.6)	−0.1 (−1.7, 0.6)	−0.6 (−2.2, 0.1)
Nutritional status, *n* (%)					
Severely undernourished ^2^	34 (39.1)	79 (46.5)	29 (17.7)	2 (7.7)	4 (14.3)
Undernourished ^2^	11 (12.6)	30 (17.6)	19 (11.6)	2 (7.7)	3 (10.7)
Normal ^2^	38 (43.7)	55 (32.4)	103 (62.8)	21 (80.8)	21 (75.0)
Overnourished ^2^	4 (4.6)	6 (3.5)	13 (7.9)	1 (3.8)	0 (0.0)
**Ghana**					
*n*	376	439	438	212	191
Mean (SD)	−1.5 (2.2)	−2.2 (2.8)	−0.4 (3.0)	−0.9 (3.2)	−1.6 (1.8)
Median (IQR)	−1.5 (−2.9, -0.1)	−2.1 (−3.7, −0.8)	−0.2 (−1.9, 1.2)	−0.5 (−2.8, 0.9)	−1.7 (−2.8, −0.4)
Nutritional status, *n* (%)					
Severely undernourished ^2^	86 (22.9)	148 (33.7)	66 (15.1)	48 (22.6)	42 (22.0)
Undernourished ^2^	60 (16.0)	77 (17.5)	38 (8.7)	21 (9.9)	37 (19.4)
Normal ^2^	215 (57.2)	194 (44.2)	261 (59.6)	108 (50.9)	108 (56.5)
Overnourished ^2^	15 (4.0)	20 (4.6)	73 (16.7)	35 (16.5)	4 (2.1)

^1^ WAZ was calculated for children aged <121 months, WHZ and MUACZ were calculated for children aged <61 months. ^2^ z-score < −3SD: severely undernourished, z-score ranged ≥ −3SD to < −2SD: undernourished, z-score ranged −2SD to +2SD: normal, and z-score > +2SD: overnourished. SD standard deviation; IQR Interquartile range.

**Table 2 nutrients-13-03676-t002:** Nutritional status of children with CP registered into the GLM CPR according to their sociodemographic characteristics.

Characteristics	Underweight (WAZ < −2SD) ^1^, *n* (%)				Stunted (HAZ < −2SD), *n* (%)				Thin (BAZ < −2SD), *n* (%)			
	Population-Based			Institution-Based	Population-Based			Institution-Based	Population-Based			Institution-Based
	Bangladesh	Indonesia	Nepal	Ghana	Bangladesh	Indonesia	Nepal	Ghana	Bangladesh	Indonesia	Nepal	Ghana
*n*	1431	71	45	146	2161	119	109	194	961	46	48	104
**Age**												
0–4	683 (48.1)	30 (42.3)	9 (20.0)	87 (61.7)	770 (35.8)	33 (27.7)	12 (11.0)	95 (43.8)	256 (26.9)	9 (19.6)	5 (10.6)	62 (63.3)
5–9	722 (50.8)	40 (56.3)	35 (77.8)	54 (38.3)	726 (33.8)	41 (34.5)	38 (34.9)	64 (29.5)	331 (34.8)	12 (26.1)	16 (34.0)	27 (27.6)
10–14	14 (1.0)	1 (1.4)	1 (2.2)	0 (0.0)	438 (20.4)	30 (25.2)	38 (34.9)	40 (18.4)	260 (27.3)	18 (39.1)	14 (29.8)	9 (9.2)
15 and above	n/a	n/a	n/a	n/a	215 (10.0)	15 (12.6)	21 (19.3)	18 (8.3)	104 (10.9)	7 (15.2)	12 (25.5)	0 (0.0)
*p-*value	**0.025**	0.05	0.15	0.20	**<0.001**	0.41	0.49	**<0.001**	**<0.001**	0.22	0.58	**0.001**
Sex of the child												
Male	871 (60.9)	38 (53.5)	23 (51.1)	92 (63.4)	1312 (60.7)	67 (56.3)	62 (56.9)	138 (61.6)	602 (62.6)	25 (54.3)	29 (60.4)	70 (67.3)
Female	560 (39.1)	33 (46.5)	22 (48.9)	53 (36.6)	849 (39.3)	52 (43.7)	47 (43.1)	86 (38.4)	359 (37.4)	21 (45.7)	19 (39.6)	34 (32.7)
*p-*value	0.70	0.22	**0.03**	0.09	0.07	0.64	**0.03**	0.10	0.32	0.79	0.75	**0.02**
**Maternal education level (formal schooling)**												
None	379 (26.5)	5 (7.0)	11 (25.0)	40 (27.8)	657 (30.5)	15 (12.6)	43 (41.7)	79 (35.3)	328 (34.2)	8 (17.4)	22 (45.8)	28 (27.5)
Primary	602 (42.2)	36 (50.7)	17 (38.6)	57 (39.6)	876 (40.6)	59 (49.6)	34 (33.0)	78 (34.8)	381 (39.8)	21 (45.7)	11 (22.9)	39 (38.2)
Secondary	381 (26.7)	11 (15.5)	0 (0.0)	28 (19.4)	526 (24.4)	19 (16.0)	5 (4.9)	33 (14.7)	205 (21.4)	7 (15.2)	1 (2.1)	22 (21.6)
≥Higher secondary	66 (4.6)	19 (26.8)	16 (36.4)	19 (13.2)	96 (4.5)	26 (21.8)	21 (20.4)	34 (15.2)	44 (4.6)	10 (21.7)	14 (29.2)	13 (12.7)
*p-*value	**<0.001**	0.17	0.15	**0.01**	**0.005**	0.53	0.14	**0.01**	**0.01**	0.87	0.39	0.22
**Paternal education level (formal schooling)**												
None	510 (35.9)	10 (14.7)	9 (20.5)	24 (17.1)	846 (39.5)	17 (14.9)	21 (21.0)	41 (18.4)	415 (43.4)	8 (18.6)	7 (15.2)	17 (17.0)
Primary	496 (35.0)	32 (47.1)	12 (27.3)	47 (33.6)	686 (32.1)	57 (50.0)	25 (25.0)	83 (37.2)	299 (31.2)	18 (41.9)	12 (26.1)	28 (28.0)
Secondary	296 (20.9)	10 (14.7)	5 (11.4)	38 (27.1)	438 (20.5)	13 (11.4)	19 (19.0)	47 (21.1)	173 (18.1)	5 (11.6)	9 (19.6)	34 (34.0)
≥Higher secondary	117 (8.2)	16 (23.5)	18 (40.9)	31 (22.1)	170 (7.9)	27 (23.7)	35 (35.0)	52 (23.3)	70 (7.3)	12 (27.9)	18 (39.1)	21 (21.0)
*p-*value	**0.005**	0.74	0.49	**0.02**	**0.04**	0.88	0.93	**0.02**	**0.01**	0.96	0.82	**0.004**
**Maternal occupation (involved in any income generating activities)**												
Yes	171 (12.0)	54 (79.4)	27 (64.3)	120 (82.8)	252 (11.7)	97 (83.6)	63 (63.6)	186 (83.0)	117 (12.2)	41 (91.1)	26 (59.1)	88 (84.6)
No	1252 (88.0)	14 (20.6)	15 (35.7)	25 (17.2)	1896 (88.3)	19 (16.4)	36 (36.4)	38 (17.0)	842 (87.8)	4 (8.9)	18 (40.9)	16 (15.4)
*p-*value ^2^	0.81	0.54	0.37	0.30	0.52	0.30	0.07	0.08	0.94	0.19 ^3^	0.84	0.72
**Paternal occupation (involved in any income generating activities)**												
Yes	1399 (99.0)	66 (100.0)	40 (93.0)	135 (94.4)	2110 (99.2)	112 (100.0)	98 (97.0)	213 (96.4)	938 (98.7)	45 (100.0)	45 (97.8)	99 (96.1)
No	14 (1.0)	0 (0.0)	3 (7.0)	8 (5.6)	16 (0.8)	0 (0.0)	3 (3.0)	8 (3.6)	12 (1.3)	0 (0.0)	1 (2.2)	4 (3.9)
*p-*value ^2^	0.06	n/a	0.16 ^3^	0.11	0.55	n/a	0.55 ^3^	0.20	**0.01**	n/a	0.48 ^3^	0.31 ^3^
**Source of drinking water**												
Tubewell/Well/Borhole	1402 (98.2)	29 (40.8)	1 (2.3)	42 (29.0)	2114 (98.1)	45 (37.8)	3 (2.8)	57 (26.0)	940 (98.1)	15 (32.6)	2 (4.2)	28 (27.2)
Piped water/Tap water	22 (1.5)	5 (7.0)	34 (77.3)	90 (62.1)	35 (1.6)	10 (8.4)	90 (84.1)	146 (66.7)	16 (1.7)	4 (8.7)	40 (83.3)	64 (62.1)
Other sources ^4^	3 (0.2)	37 (52.1)	9 (20.5)	13 (9.0)	5 (0.2)	64 (53.8)	14 (13.1)	16 (7.3)	2 (0.2)	27 (58.7)	6 (12.5)	11 (10.7)
*p*-value ^2^	0.58 ^3^	0.70	0.98	0.34	0.50	0.50	0.94	1.00	0.80 ^3^	0.60	0.70	0.77
**Type of toilet used ^5^**												
Sanitary	705 (49.4)	45 (63.4)	40 (97.6)	106 (73.1)	1073 (49.7)	87 (73.1)	103 (98.1)	165 (74.0)	485 (50.6)	34 (73.9)	44 (95.7)	75 (72.8)
Nonsanitary	685 (48.0)	0 (0.0)	1 (2.4)	39 (26.9)	1028 (47.7)	0 (0.0)	1 (1.0)	58 (26.0)	452 (47.1)	0 (0.0)	1 (2.2)	28 (27.8)
No toilet facility	38 (2.7)	26 (36.6)	0 (0.0)	0 (0.0)	56 (2.6)	32 (26.9)	1 (1.0)	0 (0.0)	22 (2.3)	12 (26.1)	1 (2.2)	0 (0.0)
*p-*value ^2^	**0.001**	0.17	0.20	0.92 ^3^	**<0.001**	0.61 ^3^	0.40 ^3^	0.19 ^3^	1.00	0.52 ^3^	0.86 ^3^	0.39^3^
**Monthly family income (USD)**												
0 to <73.2	223 (15.7)	56 (80.0)	13 (29.5)	59 (52.7)	352 (16.4)	96 (82.1)	34 (31.5)	106 (53.5)	156 (16.4)	38 (82.6)	12 (25.0)	39 (50.0)
73.2 to <109.8	663 (46.8)	7 (10.0)	13 (29.5)	13 (11.6)	990 (46.2)	11 (9.4)	28 (25.9)	29 (14.6)	467 (49.1)	4 (8.7)	12 (25.0)	11 (14.1)
≥109.8	532 (37.5)	7 (10.0)	18 (40.9)	40 (35.7)	800 (37.3)	10 (8.5)	46 (42.6)	63 (31.8)	329 (34.6)	4 (8.7)	24 (50.0)	28 (35.9)
*p-*value ^2^	0.67	0.07	0.08	0.31	0.77	0.22	0.72	0.79	0.46	0.92 ^3^	0.79	0.38

^1^ WAZ was calculated for children aged <121 months. ^2^ Chi-square test (two-sided). ^3^ Fisher’s exact test (two-sided). ^4^ Others include ponds, river, spring, rainwater, and bottled water. ^5^ Defined following the DHS categories. n/a = not applicable. the bold fonts indicate statitical significance.

**Table 3 nutrients-13-03676-t003:** Nutritional status of the participating children according to their clinical characteristics.

Characteristics	Underweight (WAZ < −2SD) ^1^, *n* (%)				Stunted (HAZ < −2SD), *n* (%)				Thin (BAZ < −2SD), *n* (%)			
	Population-Based			Institution-Based	Population-Based			Institution-Based	Population-Based			Institution-Based
	Bangladesh	Indonesia	Nepal	Ghana	Bangladesh	Indonesia	Nepal	Ghana	Bangladesh	Indonesia	Nepal	Ghana
*n*	1431	71	45	146	2161	119	109	225	961	46	48	104
**Predominant motor type**												
Spastic	1113 (77.8)	52 (73.2)	39 (86.7)	104 (71.2)	1744 (80.7)	98 (82.4)	89 (81.7)	171 (76.0)	778 (81.0)	41 (89.1)	40 (83.3)	69 (66.3)
Dyskinesia	87 (6.1)	11 (15.5)	0 (0.0)	12 (8.2)	138 (6.4)	10 (8.4)	4 (3.7)	9 (4.0)	66 (6.9)	3 (6.5)	2 (4.2)	9 (8.7)
Ataxia	48 (3.4)	0 (0.0)	1 (2.2)	15 (10.3)	62 (2.9)	1 (0.8)	2 (1.8)	23 (10.2)	26 (2.7)	1 (2.2)	3 (6.3)	12 (11.5)
Hypotonia	183 (12.8)	8 (11.3)	5 (11.1)	15 (10.3)	216 (10.0)	10 (8.4)	14 (12.8)	22 (9.8)	91 (9.5)	1 (2.2)	3 (6.3)	14 (13.5)
*p*-value ^2^	0.32	0.54 ^3^	**0.02** ^3^	0.81	0.19	0.31^3^	**<0.001** ^3^	**0.01**	0.34	0.15 ^3^	0.42 ^3^	0.50
**Spastic Topography**												
Mono/hemiplegia	234 (21.0)	8 (15.4)	11 (28.2)	17 (16.3)	346 (19.8)	14 (14.3)	26 (29.2)	27 (15.8)	178 (22.9)	6 (14.6)	12 (30.0)	16 (23.2)
Diplegia	194 (17.4)	8 (15.4)	5 (12.8)	25 (24.0)	298 (17.1)	19 (19.4)	10 (11.2)	35 (20.5)	128 (16.5)	8 (19.5)	3 (7.5)	16 (23.2)
Tri/quadriplegia	685 (61.5)	36 (69.2)	23 (59.0)	62 (59.6)	1100 (63.1)	65 (66.3)	53 (59.6)	109 (63.7)	472 (60.7)	27 (65.9)	25 (7.5)	37 (53.6)
*p*-value ^2^	**<0.001**	0.91	0.53	0.52	**<0.001**	0.65	**0.002**	**0.003**	**<0.001**	0.97	0.18 ^3^	0.65
**GMFCS level**												
I–II	272 (19.1)	7 (9.9)	13 (28.9)	48 (33.8)	406 (18.9)	19 (16.0)	37 (33.9)	63 (28.3)	227 (23.8)	8 (17.4)	15 (31.3)	44 (43.1)
III–V	1153 (80.9)	64 (90.1)	32 (71.1)	94 (66.2)	1746 (81.1)	100 (84.0)	72 (66.1)	160 (71.7)	727 (76.2)	38 (82.6)	33 (68.8)	58 (56.9)
*p*-value ^2^	**<0.001**	**0.01**	**0.01**	**0.02**	**<0.001**	0.33	**<0.001**	**<0.001**	**0.002**	0.61	**0.01**	0.82
**Associated impairment**												
None	215 (15.0)	9 (12.7)	0 (0.0)	38 (26.0)	336 (15.5)	15 (12.6)	2 (1.8)	37 (16.4)	140 (14.6)	4 (8.7)	0 (0.0)	32 (30.8)
One	400 (28.0)	26 (36.6)	15 (33.3)	48 (32.9)	599 (27.7)	52 (43.7)	33 (30.3)	100 (44.4)	288 (30.0)	20 (43.5)	15 (31.3)	34 (32.7)
Two and more	815 (57.0)	36 (50.7)	30 (66.7)	60 (41.1)	1226 (56.7)	52 (43.7)	74 (67.9)	88 (39.1)	532 (55.4)	22 (47.8)	33 (68.8)	38 (36.5)
*p*-value ^2^	**<0.001**	0.10	0.52 ^3^	**0.004**	**<0.001**	0.99	0.82 ^3^	**<0.001**	**0.002**	0.31 ^3^	0.61 ^3^	0.33
**Dysphagia**												
No	958 (67.0)	43 (61.4)	24 (54.5)	n/a	1457 (67.6)	73 (61.9)	67 (61.5)	n/a	650 (68.0)	25 (54.3)	29 (61.7)	n/a
Yes	472 (33.0)	27 (38.6)	20 (45.5)	n/a	697 (32.4)	45 (38.1)	42 (38.5)	n/a	306 (32.0)	21 (45.7)	18 (38.3)	n/a
*p*-value ^2^	**<0.001**	0.37	0.43	n/a	**<0.001**	0.47	0.11	n/a	**0.01**	0.23	0.48	n/a
**Gastro-esophageal reflux**												
No	1088 (76.1)	64 (92.8)	24 (64.9)	n/a	1642 (76.3)	107 (93.0)	67 (76.1)	n/a	749 (78.4)	37 (86.0)	30 (75.0)	n/a
Yes	341 (23.9)	5 (7.2)	13 (35.1)	n/a	511 (23.7)	8 (7.0)	21 (23.9)	n/a	206 (21.6)	6 (14.0)	10 (25.0)	n/a
*p*-value ^2^	**0.001**	0.57	0.11	n/a	**<0.001**	0.42	**0.04**	n/a	0.41	**0.04**	0.34	n/a

^1^ WAZ was calculated for children aged <121 months. ^2^ Chi-square test (two-sided). ^3^ Fisher’s exact test (two-sided). n/a not applicableWe also observed that the majority of children with spastic tri/quadriplegia had GMFCS levels III–V (*p* < 0.001 for all) and two or more associated impairments in all four countries (*p* < 0.001, *p* = 0.20, *p* = 0.04, and *p* < 0.001 in Bangladesh, Indonesia, Nepal, and Ghana). Moreover, the presence of dysphagia was comparatively higher among children with GMFCS levels III−V than GMFCS levels I−II in Bangladesh (*p* < 0.001) and Nepal (*p* = 0.001). Furthermore, a positive relationship between dysphagia and gastroesophageal reflux was observed among children with CP in Bangladesh (*p* < 0.001) and Nepal (*p* < 0.001).

**Table 4 nutrients-13-03676-t004:** Predictors of underweight (WAZ < −2SD) among children with CP registered in GLM CPR (adjusted odds ratio—aOR).

Predictors	Bangladesh		Indonesia		Nepal		Ghana	
	aOR (95% CI)	*p*-Value	aOR (95% CI)	*p*-Value	aOR (95% CI)	*p*-Value	aOR (95% CI)	*p*-Value
**Age (years)**								
0–4	Ref		Ref		n/a	n/a	n/a	n/a
5–9	1.4 (1.1,1.8)	**0.02**	14.9 (1.4, 162.0)	**0.03**	n/a	n/a	n/a	n/a
10–14	1.2 (0.4, 3.8)	0.74	-	1.00				
**Sex of the child**								
Male	n/a	n/a	n/a	n/a	Ref		n/a	n/a
Female	n/a	n/a	n/a	n/a	2.5 (0.9, 6.5)	0.06	n/a	n/a
**Maternal educational level (formal schooling)**								
None	2.8 (1.5, 5.4)	**0.002**	n/a	n/a	n/a	n/a	2.1 (1.1, 4.3)	**0.03**
Primary	2.6 (1.4, 4.8)	**0.002**	n/a	n/a	n/a	n/a	2.9 (1.5, 5.7)	**0.002**
Secondary	2.1 (1.2, 3.8)	**0.01**	n/a	n/a	n/a	n/a	1.7 (0.8, 3.5)	0.16
≥Higher secondary	Ref		n/a	n/a	n/a	n/a	Ref	
**Paternal educational level (formal schooling)**								
None	0.8 (0.5, 1.5)	0.58	n/a	n/a	n/a	n/a	n/a	n/a
Primary	1.1 (0.6, 1.8)	0.76	n/a	n/a	n/a	n/a	n/a	n/a
Secondary	0.8 (0.5, 1.3)	0.31	n/a	n/a	n/a	n/a	n/a	n/a
≥Higher secondary	Ref		n/a	n/a	n/a	n/a	n/a	n/a
**Predominant motor type**								
Spastic	n/a	n/a	n/a	n/a	Ref		n/a	n/a
Dyskinesia	n/a	n/a	n/a	n/a	n/a	n/a	n/a	n/a
Ataxia	n/a	n/a	n/a	n/a	n/a	n/a	n/a	n/a
Hypotonia	n/a	n/a	n/a	n/a	n/a	n/a	n/a	n/a
**Spastic Topography**								
Mono/hemiplegia	Ref		n/a	n/a	n/a	n/a	n/a	n/a
Diplegia	1.7 (1.2, 2.4)	**0.01**	n/a	n/a	n/a	n/a	n/a	n/a
Tri/quadriplegia	3.1 (2.2, 4.5)	**<0.001**	n/a	n/a	n/a	n/a	n/a	n/a
**GMFCS level**								
I–II	Ref		Ref		Ref		Ref	
III–V	1.8 (1.3, 2.5)	**<0.001**	14.9 (2.2, 102.3)	0.01	3.2 (1.2, 8.1)	**0.01**	1.3 (0.8, 2.2)	0.29
**Associated impairment**								
None	Ref		Ref		n/a	n/a	Ref	
One	1.6 (1.1, 2.3)	**0.01**	n/a	n/a	n/a	n/a	0.9 (0.5, 1.7)	0.88
Two or more	1.6 (1.1, 2.2)	**0.01**	n/a	n/a	n/a	n/a	1.9 (1.0, 3.5)	0.05
**Dysphagia**								
No	Ref		n/a	n/a	n/a	n/a	n/a	n/a
Yes	1.0 (0.7, 1.5)	0.95	n/a	n/a	n/a	n/a	n/a	n/a
**Gastroesophageal reflux**								
No	Ref		n/a	n/a	n/a	n/a	n/a	n/a
Yes	1.2 (0.8, 1.8)	0.41	n/a	n/a	n/a	n/a	n/a	n/a

n/a not applicable.

**Table 5 nutrients-13-03676-t005:** Predictors of stunting (HAZ < −2SD) among children with CP registered in the GLM CPR (adjusted odds ratio—aOR).

Predictors	Bangladesh		Indonesia		Nepal		Ghana	
	aOR (95% CI)	*p*-Value	aOR (95% CI)	*p*-Value	aOR (95% CI)	*p*-Value	aOR (95% CI)	*p*-Value
**Age group (years)**								
0–4	Ref		n/a	n/a	n/a	n/a	Ref	
5–9	0.7 (0.5, 1.0)	**0.02**	n/a	n/a	n/a	n/a	0.7 (0.5, 1.2)	0.25
10–14	0.5 (0.3, 0.6)	**<0.001**	n/a	n/a	n/a	n/a	2.8 (1.3, 6.4)	**0.01**
15 and above	1.5 (0.9, 2.6)	0.09	n/a	n/a	n/a	n/a	9.0 (1.9, 42.4)	**0.01**
**Sex**								
Female	n/a	n/a	n/a	n/a	2.3 (1.0, 5.1)	0.05	n/a	n/a
Male	n/a	n/a	n/a	n/a	Ref		n/a	n/a
**Maternal educational level (formal schooling)**								
None	2.1 (1.2, 3.5)	**0.01**	n/a	n/a	n/a	n/a	1.7 (0.7, 4.3)	0.26
Primary	2.5 (1.5, 4.1)	**<0.001**	n/a	n/a	n/a	n/a	1.3 (0.6, 3.1)	0.47
Secondary	2.0 (1.2, 3.4)	**0.01**	n/a	n/a	n/a	n/a	0.6 (0.3, 1.4)	0.25
≥Higher secondary	Ref		n/a	n/a	n/a	n/a	Ref	
**Paternal educational level (formal schooling)**								
None	n/a	n/a	n/a	n/a	n/a	n/a	0.5 (0.2, 1.3)	0.18
Primary	n/a	n/a	n/a	n/a	n/a	n/a	0.9 (0.4, 2.0)	0.81
Secondary	n/a	n/a	n/a	n/a	n/a	n/a	1.0 (0.5, 2.1)	0.96
≥Higher secondary	Ref		n/a	n/a	n/a	n/a	Ref	
**Predominant motor type**								
Spastic	n/a	n/a	n/a	n/a	Ref		n/a	n/a
Dyskinesia	n/a	n/a	n/a	n/a	1.0 (0.2, 5.2)	0.97	n/a	n/a
Ataxia	n/a	n/a	n/a	n/a	0.1 (0.0, 0.4)	**0.002**	n/a	n/a
Hypotonia	n/a	n/a	n/a	n/a	2.4 (0.5, 11.9)	0.28	n/a	n/a
**Spastic topography**								
Mono/hemiplegia	Ref		n/a	n/a	n/a	n/a	n/a	n/a
Diplegia	1.8 (1.3, 2.5)	**<0.001**	n/a	n/a	n/a	n/a	n/a	n/a
Tri/quadriplegia	4.7 (3.4, 6.5)	**<0.001**	n/a	n/a	n/a	n/a	n/a	n/a
**GMFCS level**								
I–II	Ref		n/a	n/a	Ref		Ref	
III–V	2.6 (2.0, 3.4)	**<0.001**	n/a	n/a	3.4 (1.6, 7.1)	**0.001**	1.6 (1.0, 2.8)	0.06
**Associated impairment**								
None	Ref		n/a	n/a	n/a	n/a	Ref	
One	1.0 (0.8, 1.4)	0.79	n/a	n/a	n/a	n/a	2.7 (1.4, 5.1)	**0.002**
Two and more	1.1 (0.8, 1.6)	0.40	n/a	n/a	n/a	n/a	3.0 (1.5, 5.8)	**0.002**
**Dysphagia**								
No	Ref		n/a	n/a	n/a	n/a	n/a	n/a
Yes	1.1 (0.7, 1.6)	0.66	n/a	n/a	n/a	n/a	n/a	n/a
**Gastroesophageal reflux**								
No	Ref		n/a	n/a	n/a	n/a	n/a	n/a
Yes	1.3 (0.9, 2.1)	0.17	n/a	n/a	n/a	n/a	n/a	n/a

n/a not applicable.

**Table 6 nutrients-13-03676-t006:** Predictors of thinness (BAZ < −2SD) among children with CP registered in the GLM CPR (adjusted odds ratio—aOR).

Predictors	Bangladesh		Indonesia		Nepal		Ghana	
	aOR (95% CI)	*p*-Value	aOR (95% CI)	*p*-Value	aOR (95% CI)	*p*-Value	aOR (95% CI)	*p*-Value
**Age group (years)**								
0–4	Ref		n/a	n/a	n/a	n/a	Ref	
5–9	1.3 (1.1, 1.7)	**0.01**	n/a	n/a	n/a	n/a	0.4 (0.2, 0.7)	**0.001**
10–14	1.7 (1.4, 2.3)	**<0.001**	n/a	n/a	n/a	n/a	0.5 (0.2, 1.1	0.13
15 and above	1.6 (1.1, 2.3)	**0.01**	n/a	n/a	n/a	n/a	n/a	1.00
**Sex of the child**								
Male	n/a	n/a	n/a	n/a	n/a	n/a	Ref	
Female	n/a	n/a	n/a	n/a	n/a	n/a	0.5 (0.3, 0.9)	**0.02**
**Maternal educational level (formal schooling)**								
No formal schooling	1.2 (0.7, 2.1)	0.48	n/a	n/a	n/a	n/a	n/a	n/a
Primary	1.1 (0.7, 1.9)	0.67	n/a	n/a	n/a	n/a	n/a	n/a
Secondary	0.9 (0.6, 1.6)	0.80	n/a	n/a	n/a	n/a	n/a	n/a
Higher secondary and above	Ref		n/a	n/a	n/a	n/a	n/a	n/a
**Paternal educational level (formal schooling)**								
No formal schooling	1.2 (0.8, 1.8)	0.46	n/a	n/a	n/a	n/a	0.9 (0.4, 2.1)	0.91
Primary	1.0 (0.7, 1.6)	0.81	n/a	n/a	n/a	n/a	1.2 (0.6, 2.4)	0.59
Secondary	0.9 (0.6, 1.4)	0.62	n/a	n/a	n/a	n/a	2.6 (1.3, 5.3)	**0.01**
Higher secondary and above	Ref		n/a	n/a	n/a	n/a	Ref	
**Spastic topography**								
Mono/hemiplegia	Ref		n/a	n/a	n/a	n/a	n/a	n/a
Diplegia	1.3 (1.0, 1.7)	1.00	n/a	n/a	n/a	n/a	n/a	n/a
Tri/quadriplegia	1.5 (1.2, 2.0)	**0.002**	n/a	n/a	n/a	n/a	n/a	n/a
**GMFCS level**								
I–II	Ref		n/a	n/a	Ref		n/a	n/a
III–V	1.0 (0.8, 1.4)	0.70	n/a	n/a	2.4 (1.2, 5.0)	**0.02**	n/a	n/a
**Associated impairment**								
None	Ref		n/a	n/a	n/a	n/a	n/a	n/a
One	1.3 (1.0, 1.8)	**0.04**	n/a	n/a	n/a	n/a	n/a	n/a
Two and more	1.4 (1.1, 1.9)	**0.01**	n/a	n/a	n/a	n/a	n/a	n/a
**Dysphagia**								
No	Ref		n/a	n/a	n/a	n/a	n/a	n/a
Yes	1.1 (0.9, 1.4)	0.28	n/a	n/a	n/a	n/a	n/a	n/a

n/a not applicable.

## Data Availability

The data presented in this study are available on request from the corresponding author. The data are not publicly available due to privacy/ethical restrictions.

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
