# Peer review of "Epidemiology of Malnutrition among Children with Cerebral Palsy in Low- and Middle-Income Countries: Findings from the Global LMIC CP Register"

_nutrients, 2021, doi:10.3390/nu13113676_

Round 1

Reviewer 1 Report

Introduction:

- I suggest to describe the areas were CP is prevalent to justify the 4 selected countries selected.

Methods:

In order to better understand the methodology:

  • To define the type of interview done to the participants (in person/ on-line, duration, who answer,...)
  • It would be interesting to analyse statistical correlation among health indications but also with social factors. Please review the possible correlations and explain whether they are needed to include in results.

Results:

- I suggest to review the format of the tables, for instance percentages are not always needed to put it since no information is added.

Discussion:

- I would appreciate any literature reference for the explanation of the differences between countries even the cultural or social aspects.

Conclusions:

  • The idea is not well explained, please define the main conclusion in relation with your objective and thereafter the limitations that you mentioned.

Author Response

We would like to thank the respected reviewer for the constructive suggestions and helpful comments on our manuscript titled “Epidemiology of malnutrition among children with cerebral palsy in low- and middle-income countries: findings from the Global LMIC CP Register”.

Please see below our point-by-point response to the reviewer’s comments;

Point 1: Introduction

I suggest to describe the areas were CP is prevalent to justify the 4 selected countries selected.

Response 1: Thank you for the valuable suggestion. We have added a few sentences in the introduction section as suggested. Please see line 51-52, 68-69, 72-74 of the edited manuscript.

Point 2: Methods

In order to better understand the methodology:

  • To define the type of interview done to the participants (in person/ on-line, duration, who answer,...)

Response 2: Thank you for the valuable suggestion. Data were collected in person. We have now added this information in the methodology. Please see line 108-113 of the edited manuscript.

Point 3: Results

  • It would be interesting to analyse statistical correlation among health indications but also with social factors. Please review the possible correlations and explain whether they are needed to include in results.

Response 3: Thank you for the valuable suggestion. We have explored the statistical correlation between different socio-demographic and clinical factors and observed a positive relationship between maternal and paternal education, as well as between GMFCS level, spastic topography, associated impairments, dysphagia and gastro-esophageal reflux among children with CP in different LMICs included in the GLM CPR. We have now added that information in the results and discussions accordingly. Please see line 228-230, 268-274, 346-355 of the edited manuscript.

Point 4: Results

I suggest to review the format of the tables, for instance percentages are not always needed to put it since no information is added.

Response 4: Thank you for the thoughtful suggestion. We have now changed the format in Table 1 as suggested.

Point 5: Discussion

I would appreciate any literature reference for the explanation of the differences between countries even the cultural or social aspects.

Response 5: Thank you for the valuable suggestion. We have edited the discussion section accordingly. Please see line 379-390 of the edited manuscript.

Point 6: Conclusions

  • The idea is not well explained, please define the main conclusion in relation with your objective and thereafter the limitations that you mentioned.

Response 6: Thank you for the valuable suggestion. We have revised the conclusion section accordingly. Please see line 410-419, 425 of the edited manuscript.

Thank you. 

Reviewer 2 Report

Thank you for the opportunity to review the manuscript “Epidemiology of malnutrition among children with cerebral  palsy in low- and middle-income countries: findings from the  Global LMIC CP Register” The authors presented the results of the epidemiological study of malnutrition among children with cerebral palsy (CP) in low-and-middle-income countries (LMICs).  
The research was carefully designed, the manuscript is well written and clear. The conclusions are consistent with the results presented and they all address the main question of the research. Scientifically predictable results, but can be a good basis for applying for funding for educational programs. Research results important for public health.

Author Response

We would like to thank the respected reviewer for the constructive suggestions and helpful comments on our manuscript titled “Epidemiology of malnutrition among children with cerebral palsy in low- and middle-income countries: findings from the Global LMIC CP Register”.

Please see below our point-by-point response to the reviewer’s comments;

Point 1: Thank you for the opportunity to review the manuscript “Epidemiology of malnutrition among children with cerebral palsy in low- and middle-income countries: findings from the Global LMIC CP Register” The authors presented the results of the epidemiological study of malnutrition among children with cerebral palsy (CP) in low-and-middle-income countries (LMICs). The research was carefully designed, the manuscript is well written and clear. The conclusions are consistent with the results presented and they all address the main question of the research. Scientifically predictable results, but can be a good basis for applying for funding for educational programs. Research results important for public health.

Response 1: Thank you very much for the positive feedback.